# Targeted Disruption of Bone Marrow Stromal Cell-Derived Gremlin1 Limits Multiple Myeloma Disease Progression In Vivo

**DOI:** 10.3390/cancers12082149

**Published:** 2020-08-03

**Authors:** Kimberley C. Clark, Duncan R. Hewett, Vasilios Panagopoulos, Natalya Plakhova, Khatora S. Opperman, Alanah L. Bradey, Krzysztof M. Mrozik, Kate Vandyke, Siddhartha Mukherjee, Gareth C.G. Davies, Daniel L. Worthley, Andrew C.W. Zannettino

**Affiliations:** 1Myeloma Research Laboratory, Adelaide Medical School, Faculty of Health and Medical Sciences, University of Adelaide, Adelaide, SA 5000, Australia; kimberley.clark@monash.edu (K.C.C.); duncan.hewett@adelaide.edu.au (D.R.H.); vasilios.panagopoulos@adelaide.edu.au (V.P.); Natalya.plakhova@adelaide.edu (N.P.); khatora.opperman@adelaide.edu.au (K.S.O.); alanah.bradey@adelaide.edu.au (A.L.B.); Krzysztof.mrozik@adelaide.edu.au (K.M.M.); kate.vandyke@adelaide.edu.au (K.V.); 2Precision Medicine Theme, South Australian Health and Medical Research Institute, Adelaide, SA 5000, Australia; dan.worthley@sahmri.com; 3Department of Medicine, Columbia University Medical Center, New York City, NY 10032, USA; sm3252@cumc.columbia.edu; 4UCB (Union Chimique Belge) Pharma, Slough SL1 3WE, UK; gareth.davies@ucb.com; 5Central Adelaide Local Health Network, Adelaide, SA 5000, Australia

**Keywords:** multiple myeloma, Grem1, bone marrow stromal cells, Gremlin1

## Abstract

In most instances, multiple myeloma (MM) plasma cells (PCs) are reliant on factors made by cells of the bone marrow (BM) stroma for their survival and growth. To date, the nature and cellular composition of the BM tumor microenvironment and the critical factors which drive tumor progression remain imprecisely defined. Our studies show that Gremlin1 (Grem1), a highly conserved protein, which is abundantly secreted by a subset of BM mesenchymal stromal cells, plays a critical role in MM disease development. Analysis of human and mouse BM stromal samples by quantitative PCR showed that *GREM1*/*Grem1* expression was significantly higher in the MM tumor-bearing cohorts compared to healthy controls (*p* < 0.05, Mann–Whitney test). Additionally, BM-stromal cells cultured with 5TGM1 MM PC line expressed significantly higher levels of *Grem1*, compared to stromal cells alone (*p* < 0.01, *t*-test), suggesting that MM PCs promote increased *Grem1* expression in stromal cells. Furthermore, the proliferation of 5TGM1 MM PCs was found to be significantly increased when co-cultured with *Grem1*-overexpressing stromal cells (*p* < 0.01, *t*-test). To examine the role of Grem1 in MM disease in vivo, we utilized the 5TGM1/KaLwRij mouse model of MM. Our studies showed that, compared to immunoglobulin G (IgG) control antibody-treated mice, mice treated with an anti-Grem1 neutralizing antibody had a decrease in MM tumor burden of up to 81.2% (*p* < 0.05, two-way ANOVA). The studies presented here demonstrate, for the first time, a novel positive feedback loop between MM PCs and BM stroma, and that inhibiting this vicious cycle with a neutralizing antibody can dramatically reduce tumor burden in a preclinical mouse model of MM.

## 1. Introduction

Multiple myeloma (MM) is a hematological malignancy characterized by the uncontrolled proliferation of antibody-producing plasma cells (PC) within the bone marrow (BM) [1]. MM is defined by the presence of 10% or more clonal PC in the BM and one or more myeloma-defining event(s) [1]. Myeloma-defining events include evidence of end-organ damage, such as hypercalcemia, renal insufficiency, anemia, and bone lesions, known as the CRAB criteria [2]. While recent treatment advances have improved the median overall survival for MM to approximately 6 years [3], the majority of patients relapse and MM remains a largely incurable disease [4]. Almost all cases of MM are preceded by monoclonal gammopathy of undetermined significance (MGUS), a benign clonal PC proliferation characterized by less than 10% PCs in the BM and the absence of end-organ damage [5].

The initiating events in the development of MGUS have been shown to occur in post-germinal center B cell via a primary cytogenetic event, including chromosomal translocations involving the immunoglobulin heavy-chain gene [6]. While malignant transformation and MM disease progression are believed to occur due to the accumulation of secondary “genetic hits”, recent studies from our laboratory [7] and others [8,9] observed that many of the chromosomal abnormalities and genetic lesions identified in MM PCs are also found at the MGUS stage, suggesting that PC-extrinsic factors may be involved in driving the progression from asymptomatic MGUS to malignant MM. Like normal PCs, MM PCs are home to C-X-C motif chemokine 12 (CXCL12)-rich regions of the BM microenvironment, where they are exposed to a variety of extracellular matrix proteins, BM stromal cells, and other juxtaposed cells [10]. While it is well established that the BM microenvironment is critical for MM tumor growth, it remains to be elucidated which stromal cells and factors are required for the continued growth, spread, and survival of the MM PC [11,12].

To this end, MM PC are thought to hijack niches that normally support hematopoietic stem cell self-renewal and multi-lineage differentiation [13]. The hematopoietic stem cell niche is defined by supportive signals that are derived, in large part, from skeletal or mesenchymal stem cells or their progeny [13]. Currently, there are two recognized populations of skeletal stem cells, namely the perisinusoidal mesenchymal stem cells, previously described by our group [14] and others [15,16], and a rare, discrete population of osteochondroreticular stem cells, recently described by Worthley et al. [17]. Osteochondroreticular stem cells display a capacity for extensive self-renewal and differentiate into osteoblasts, chondrocytes, and reticular marrow stromal cells, but not adipocytes. While traditional mesenchymal stem cells are located around the marrow sinusoids, osteochondroreticular stem cells are predominantly found within the metaphyseal and endosteal regions of bone, sites commonly associated with MM PC tumor growth [14,17,18].

Osteochondroreticular stem cells are defined by their expression of Gremlin1 (Grem1), a highly conserved, 184 amino acid cysteine-knot secreted protein, belonging to the differential screening-selected gene in neuroblastoma (DAN) family of glycoproteins. Grem1 has been shown to be a potent bone morphogenic protein (BMP)-2, -4, -7 antagonist and vascular endothelial growth factor receptor-2 (VEGFR-2) agonist [19,20]. Grem1 is expressed by cells of the periepithelial intestinal mesenchymal sheath [17] and is functionally implicated in driving gastrointestinal carcinogenesis [21]. Stromal-derived Grem1 has also been implicated within the unique tumor microenvironment of other cancers including mesothelioma [22], pancreatic neuroendocrine tumors [23], mammary [24], uterine [25], cervical [26], and kidney [27] tumors. Furthermore, Grem1 has been found to be one of the most upregulated genes in the tumor microenvironment [28]. Consistent with its pro-angiogenic properties (i.e., via binding to VEGF-R2), Chen et al. [29] showed that Grem1 expression is associated with increased microvessel density, a marker of angiogenesis, in patients with pancreatic neuroendocrine tumors. Notably, increased microvessel density is associated with poor outcomes in patients with MM [30].

These findings demonstrate a clear role for Grem1 in cancer development, especially in the microenvironmental niches that support cancer cell growth, migration, and invasion. In this study we demonstrated, for the first time, that Grem1 plays a role in MM disease progression. Importantly, analysis of patient BM trephine biopsies revealed a significant increase in *GREM1* expression in the BM microenvironment of patients with progressive MM disease. Likewise, a significant increase in BM stromal *Grem1* expression was observed in tumor-bearing bones of two commonly used preclinical mouse models of MM. Moreover, increased expression of BM-derived *Grem1* was shown to promote the proliferation of MM PC. Most notably, targeting Grem1 with a neutralizing antibody significantly reduced MM tumor burden in the 5TGM1/KaLwRij.Hsd model of MM. Taken together, these findings represent the first evidence that bone marrow stromal-derived Grem1 plays a role in MM disease progression, and that Grem1 is a viable therapeutic target for the treatment of MM.

## 2. Results

### 2.1. GREM1 Expression Is Upregulated in MM Bone Marrow Stroma

In view of the co-localization of the osteochondroreticular stem cell population with sites associated with MM PC growth [17,18], the expression of *GREM1* was analyzed in messenger RNA (mRNA) samples obtained from healthy and MM patient-derived BM stromal cell cultures. MM patient-derived BM stromal cells (*n* = 15) had significantly higher expression of *GREM1* compared to BM stroma derived from age-matched hematopoietically normal donors (*n* = 17) (*p* = 0.0004; Figure 1A). Moreover, stromal expression of *GREM1* was also significantly increased in patients with the premalignant condition, MGUS, compared to normal donors (*p* = 0.0062; Appendix A). *Grem1* expression was also investigated in the Vk*MYC and 5TGM1/KaLwRij mouse models of MM. In both of these models, injection of the Vk*MYC transplants or 5TGM1 murine MM cell line intravenously into young syngeneic C57BL6 and KaLwRij mice, respectively, results in the development of disease that recapitulates many features of human disease, including PC growth within the BM, lytic bone disease, and paraprotein production [31,32]. In this study, compact bone was isolated from healthy and MM tumor-bearing mice and analyzed for differences in *Grem1* expression. For both MM mouse models, BM stroma from tumor-bearing mice demonstrated a significant increase in the expression of *Grem1* compared to the stroma from healthy controls (*p* < 0.001; Figure 1B and *p* = 0.0493; Figure 1C). Importantly, the KaLwRij mice with the greatest 5TGM1 tumor burden, as determined by bioluminescent imaging, displayed the greatest expression of *Grem1*, with a significant positive correlation between *Grem1* expression and tumor burden observed (*p* = 0.0018, r = 0.666; Figure 1D).

To determine the effect of MM PCs on *GREM1* expression in BM stromal cells, human MM cell lines KMS-11, RPMI.8226, H929, and U266 were co-cultured in direct contact with primary human BM stromal cells derived from hematopoietically normal individuals. The KMS-11 (*p* = 0.0189) and U266 (*p* = 0.0121) cell lines stimulated an increase in *GREM1* expression in BM stroma after 72 h of co-culture, while co-culture with the cell lines RPMI-8226 (*p* > 0.05) and H929 (*p* > 0.05) did not result in changes in *GREM1* expression in the stroma (Figure 2A). Furthermore, contact co-culture of the murine MM cell line 5TGM1 adhered to the BM-derived stromal cell line OP9 resulted in a significant increase in *Grem1* expression following 72 h of co-culture compared with OP9 monocultures (*p* = 0.0200; Figure 2B). A significant change in stromal *Grem1* expression was not observed at the earlier time points (24 and 48 h) of co-culture (*p* > 0.05; Figure 2B). Co-culture was also performed in noncontact Transwell assays, however only the cell contact co-cultures exhibited statistically significant changes in OP9 stromal expression of *Grem1* (Figure 2B–C).

To explore the potential mechanism(s) underpinning this increased *Grem1* expression in co-culture conditions, we investigated the role of interleukin-6 (Il-6) in regulating Grem1. Previous studies have reported that IL-6 regulates GREM1 in a model of the fibrotic condition, systemic sclerosis [33], and it is well-established that MM PCs upregulate IL-6 in stromal cell populations [34]. However, no direct link between the cytokine and Grem1 has been reported in MM. OP9 BM stromal cells cultured in the presence of 5TGM1 MM PCs for 72 h displayed a significant increase in *Il-6* expression compared to OP9 cells cultured alone (*p* = 0.0313; Figure 2D). To investigate if this increase in *Il-6* could directly influence stromal cell expression of *Grem1*, OP9 cells were cultured with 20 ng/mL recombinant Il-6 protein for 72 h. As shown in Figure 2E, there was a significant increase in Grem1 expression in cultures exposed to Il-6 (*p* = 0.0086). To confirm the involvement of Il-6 in the upregulation of *Grem1* in stromal cells cultured in the presence of MM PCs, an Il-6 neutralizing antibody or control immunoglobulin was added to the co-culture. The addition of the Il-6 neutralizing antibody abrogated the increase in stromal *Grem1* expression when OP9 cells were cultured with 5TGM1 tumor cells (*p* < 0.05; Figure 2F). Further supporting a role for *IL6* in the regulation of *GREM1* in MM, a positive correlation between *IL-6* and *GREM1* mRNA expression was observed in both MGUS and MM patient BM stromal samples (MGUS; *p* = 0.0061, R = 0.533, and MM; *p* = 0.0458, R = 0.738); Figure 2G).

### 2.2. Increased Grem1 Expression Promotes MM PC Proliferation

To investigate the significance of increased *Grem1* expression on MM disease course, OP9 stromal cells constitutively overexpressing *Grem1* were generated through lentiviral transfection of a murine *Grem1*-expression construct (Figure 3A,B) and used in co-cultures with 5TGM1 MM PCs. The 5TGM1 cells demonstrated a significant increase in proliferation when co-cultured with the *Grem1*-overexpressing OP9 stromal cells, both in cell contact (*p* = 0.0203; Figure 3C) and Transwell assays (*p* = 0.0075; Figure 3D). These findings suggest that Grem1 may play a role in promoting the growth of MM PC within the BM microenvironment. Given that Grem1 has a primary role as a BMP-antagonist [19] and BM stromal cells are a rich source of the Grem1-targets, BMP -2, -4, and -7 [35], we hypothesized that the pro-proliferative role of Grem1 in MM occurs through antagonism of the BMP pathway. In support of this, Western blot analysis revealed that 5TGM1 MM PCs cultured in the presence of *Grem1*-overexpressing OP9 stromal cells displayed a reduction in BMP-mediated phosphorylation of the Smad pathway, compared to 5TGM1 MM PCs co-cultured with OP9-empty vector control (Appendix A).

### 2.3. Targeting Grem1 Reduces MM Tumor Burden In Vivo

Given that *Grem1* expression is increased in the MM BM microenvironment in vivo and acts to promote MM PC proliferation in vitro, we hypothesized that functional blockade of Grem1 may reduce MM tumor growth. In vitro testing of a Grem1-neutralizing antibody showed that it could effectively reverse Grem1-mediated inhibition of BMP signaling in an Inhibitor of DNA Binding-1 (ID1) reporter assay (Figure 4A). These findings were further confirmed by Western blot analyses (Figure. 4B). Notably, the addition of recombinant Grem1 to BMP2-stimulated MDA-MB-231 cells abrogated downstream BMP-signaling, as indicated by a reduction in downstream phospho-Smad levels (Figure 4B). However, the addition of anti-Grem1 antibody restored BMP-signaling (as evidenced by elevated phospho-Smad) to levels seen in the absence of recombinant Grem1 (Figure 4B). To investigate the potential for targeting Grem1 in vivo, the proliferation rate of 5TGM1 MM PCs was assessed in vitro following the addition of the anti-Grem1 antibody. The proliferation of 5TGM1 tumor cells was assessed in both MM PC-only cell suspensions and with MM PC adhered to OP9 stromal cells. No significant difference in MM PC proliferation was observed between the MM PC-only cultures (*p* > 0.05; Appendix A). However, 5TGM1 MM PC growth in co-culture with OP9 stromal cells was reduced by 17% in the presence of anti-Grem1 antibody compared to IgG control antibody (*p* < 0.0026; Appendix A).

To examine the role of Grem1 in MM tumor establishment and growth in vivo, we utilized the well-characterized 5TGM1/KaLwRij preclinical mouse model of MM [32]. Following two weeks of disease establishment, mice (*n* = 10/treatment group) were randomly assigned to two groups that displayed comparable tumor burden to receive treatment with either a neutralizing anti-Grem1 antibody or an IgG control antibody. As shown in Figure 5A, there was no significant difference in MM tumor burden at the experimental endpoint as measured by bioluminescent imaging (BLI) (*p* = 0.4399, median BLI; IgG control antibody, 1.14 × 10^7^ photons/sec (interquartile range: 1.16 × 10^6^–1.39 × 10^8^) vs. anti-Grem1 antibody, 1.40 × 10^7^ photons/sec (interquartile range: 1.12 × 10^6^–1.02 × 10^8^) (Figure 5A). Like MM PCs from patients, 5TGM1 PCs produce a monoclonal antibody (paraprotein or M-protein) that can readily be detected in peripheral blood using serum protein electrophoresis (SPEP) analysis, with quantitation of paraprotein providing an assessment of total body tumor burden. Mice treated with the anti-Grem1 antibody displayed no change in serum paraprotein levels measured by SPEP (*p* = 0.1218). While no statistically significant differences were observed in this study, there was a trend toward a reduction in tumor burden in the anti-Grem1 antibody treatment group by both BLI and SPEP (Figure 5A) that warranted further investigation.

Given these promising results, a more aggressive treatment regimen starting at an earlier timepoint was evaluated. Three days after disease initiation, mice were randomly assigned (*n* =11/treatment group) to receive treatment with either a neutralizing anti-Grem1 antibody or an IgG control antibody, with disease burden monitored by weekly BLI for the duration of the experiment. As shown in Figure 5B, anti-Grem1 therapy significantly reduced MM tumor burden in vivo at the experimental endpoint by 54.5% (*p* = 0.0012; median BLI; IgG control antibody, 1.56 × 10^7^ photons/sec (interquartile range: 1.12 × 10^6^–1.02 × 10^8^)) vs. anti-Grem1 antibody, 6.46 × 10^6^ photons/sec (interquartile range: 5.10 × 10^5^–4.65 × 10^7^, Figure 5B). Mice treated with the anti-Grem1 antibody also displayed a significant reduction in M-protein intensity compared to the control treated mice (*p* = 0.0055; Figure 5B). When mice received treatment with the anti-Grem1 therapy prior to inoculation with 5TGM1 tumor cells, an even greater reduction in tumor burden of 81.2% was observed (*p* = 0.0496; median BLI; IgG control antibody, 2.44 × 10^7^ photons/sec (interquartile range: 1.91 × 10^6^–1.91 × 10^8^) vs. anti-Grem1 antibody, 2.99 × 10^6^ photons/sec (interquartile range: 4.95 × 10^5^–3.47 × 10^7^) (Figure 5C). This was also supported by a significant reduction in M-protein intensity in mice treated with anti-Grem1 antibody (*p* = 0.0003; Figure 5C). Additionally, a 24-h in vivo BM homing assay was utilized to determine whether decreased MM PC BM homing was responsible for the greater antitumor effect observed when treatment was started prior to tumor cell inoculation (Figure 5C) compared to post tumor inoculation (Figure 5A,B). No significant difference in the homing of MM PCs to the BM (*p* > 0.05: Appendix A) or number of MM PCs remaining in circulation (*p* > 0.05: Appendix A) was observed between treatment conditions. When combined, these results indicate that Grem1 has a role in both disease establishment and progression, and represents a viable therapeutic target in MM.

Given that *GREM1* displays restricted expression in adult tissues [36], we hypothesized that the effect of anti-Grem1 antibody treatment would be specific to sites of highest Grem1 expression. Myeloma cancer cell growth would be constrained by BMPs unless a localized source of Grem1 is present, and therapeutically neutralizing Grem1 would be most effective against tumors located within Grem1-rich microenvironments. In the 5TGM1/KaLwRij model, tumor growth is commonly restricted to the skeleton and the spleen. The skeleton is reported to have moderate to high *GREM1* expression, while the spleen is reported to be negative for *GREM1* expression [36], an observation that we have independently verified by RT-PCR of splenic and compact bone RNA (Appendix A). Hence, the BLI signal in anti-Grem1 antibody and IgG control-treated mice was re-analyzed for spatially restricted BLI signal arising from either the long bones of the hindlimbs or the spleen (Figure 6A). Notably, there was a significant reduction in the mean tumor burden of the long bones of anti-Grem1 antibody-treated mice compared to control-treated mice in two of the three treatment regimens (pretreatment *p* = 0.0008, day 3 treatment *p* = 0.0022; Figure 6B). However, no significant differences in mean splenic tumor burden were observed (day 3 treatment, *p* > 0.05; Figure 6B). While the mean reduction in splenic tumor burden was reduced (pretreatment, *p* = 0.0012; Figure 6B), it was reduced far less than the mean reduction in hindlimb skeletal tumor burden.

Importantly, in vivo administration of the anti-Grem1 antibody was well tolerated, with no effects on normal hematopoiesis or gross cellular make-up of the bone marrow observed following treatment (*p* > 0.05; Figure 7) Additionally, peripheral blood analysis demonstrated no significant differences in red blood cell count, white blood cell counts, or hemoglobin levels between treatment groups (*p* > 0.05; Appendix A).

## 3. Discussion

Due to rapid bench-to-bedside translation of new therapeutic modalities, MM patient outcomes have vastly improved in recent years [4]. Despite this, the vast majority of patients inevitably relapse [4]. Furthermore, in MM patients who have received at least three lines of prior therapy and become refractory to both immunomodulatory agents and proteasome inhibitors, the mean overall survival is only 13 months [37], highlighting the urgent need for treatment advances to improve patient survival post-frontline therapy. To our knowledge, the studies presented here demonstrate, for the first time, a novel positive feedback loop between MM PCs and BM stroma involving IL-6 and Grem1. Notably, we found that inhibiting this vicious cycle with a neutralizing antibody can dramatically reduce tumor burden in a preclinical mouse model of MM.

Overexpression of *GREM1* in the tumor-supportive tissue of basal cell carcinoma, as well as carcinomas of the bladder, breast, lung, colon, and pancreas has been shown to promote tumor cell proliferation in vitro and disease progression in vivo [28,38,39]. Furthermore, aberrant *GREM1* expression in the bowel disease, hereditary mixed polyposis syndrome (HMPS), is sufficient to initiate colonic tumorigenesis [39]. Our findings are consistent with previous studies that report upregulation of *Grem1* within the tumor microenvironment, demonstrating for the first time that *Grem1* is upregulated in the BM stroma of MM patients. This was further supported in both the 5TGM1-KaLwRij and Vk*myc-C57BL/6 mouse models of MM, whereby significant increases in *Grem1* expression were observed in the bones of tumor-bearing mice compared to normal controls. Importantly, we found a positive correlation between tumor burden and *Grem1* expression. In keeping with these findings, co-culture experiments conducted with both human and mouse MM PC lines with BM-derived stroma showed increases in *GREM1/Grem1* expression within the stroma. These findings suggest that the increase in BM-derived *Grem1* expression in MM is directly driven by MM PCs. These findings are in keeping with the known interplay between MM PCs and neighboring bone marrow stromal cells, which function to provide a tumor-supportive niche within the BM [39]. The interaction and adhesion of MM PCs with the surrounding BM stroma has been shown to upregulate a number of critical pro-survival and/or anti-apoptotic pathways, including Nuclear Factor kappa-light-chain-enhancer of activated B cells (NF-κB) and Notch signaling pathways, which regulate the expression of key downstream MM supportive factors, VEGF, Insulin-like growth factor-1 (IGF-1), and IL-6 [40].

Our studies suggest that the cytokine IL-6 may be responsible for the regulation of GREM1 in MM. Paracrine and autocrine upregulation of IL-6 within the MM BM microenvironment is a common feature of MM and it is well understood that IL-6 promotes the survival and proliferation of MM tumor cells, mediated through Janus kinase/signal transducer and activator of transcription (JAK/STAT) upregulation of VEGF, Ras, Akt (Protein kinase B), and mitogen-activated protein kinase (MAPK) pathways [41]. Furthermore, studies by Uchiyama et al. [34] and others [42] have demonstrated that adhesion of MM PCs to BM stromal cells triggers stromal secretion of IL-6. This is consistent with the findings presented in this study, whereby OP9 stromal cells in direct contact with 5TGM1 MM PCs stimulate an increase in *Il-6* expression. Importantly, IL-6 has been shown to regulate the expression of *GREM1* via STAT3-dependent mechanisms in dermal fibroblasts of patients with the fibrotic condition of systemic sclerosis [33], and it is likely that a similar mechanism mediates the upregulation of stromal cell *GREM1* expression observed in our study. Furthermore, inhibition of IL-6 signaling abrogated the MM PC-mediated increase in BM stromal *Grem1* expression in MM PC stromal co-cultures. Interestingly, the human myeloma cell line (U266) that caused the highest elevation in *GREM1* expression in human BM stroma in co-culture experiments also expresses the highest levels of *IL-6* mRNA (www.keatslab.org). Taken together, these data suggest that Il-6 is critical for the upregulation of BM-derived Grem1 in MM.

Overexpression of *GREM1* in lung and colorectal cancers, as well as malignant mesothelioma, is associated with disease progression by promoting cell survival, proliferation, and invasion in vitro [21,22,28]. In contrast, increased *GREM1* expression in pancreatic neuroendocrine tumors is associated with an increase in patient progression-free survival and has been reported as a predictor of positive outcomes [29]. To examine a role for Grem1 in MM, BM stromal cells overexpressing *Grem1* were generated. MM PCs cultured with BM-stromal cells overexpressing *Grem1* exhibited up to an 80% increase in MM PC proliferation, supporting a pro-tumorigenic role for Grem1 in MM. The BMP pathway is known to inhibit MM PC proliferation and promote apoptosis and its inhibition by Grem1 represents a potential mechanism for the increase in MM PC proliferation observed [43]. In the context of MM, a number of studies have demonstrated an anti-proliferative and pro-apoptotic role for BMPs, particularly the Grem1 targets, BMP-2 and BMP-4 [43]. The findings presented here are consistent with a BMP-dependent role for Grem1 signaling, as MM cells cultured with stromal cells overexpressing *Grem1* displayed both an increase in proliferation and, subsequently, a decrease in downstream BMP signaling, as indicated by decreased phosphorylation of Smads 1, 5, and 9.

A significant setback in effectively treating MM patients relates to the diverse interpatient heterogeneity of the genetic mutations observed in the PC tumors. As such, using precision medicine to target tumor-specific dysregulated pathways will always be hampered by the small proportions of MM patients whose tumors have mutations dysregulating each specific pathway. However, in our studies, ex vivo analysis of MM patient BM trephine biopsies showed that 93% of patients had greater stromal *GREM1* expression than the median normal BM sample. As such, GREM1 may represent a microenvironmental factor that is frequently upregulated in MM patients despite their diverse tumor genetics and, therefore, may represent a suitable therapeutic target for the majority of MM patients. Given the inability to fully recapitulate the complexities of the MM BM microenvironment in vitro, the 5TGM1/KaLwRij preclinical mouse model of MM was used to further investigate a role for Grem1 in disease development and determine whether therapeutically targeting Grem1 represents a viable treatment option. Other studies have previously used neutralizing antibodies against Grem1 and reported promising results [44,45]. Kim et al. first used a Grem1-neutralizing antibody to inhibit Grem1 induced migration, invasion, and proliferation of the lung carcinoma cell line A549. A subsequent study by Ciuclan et al. demonstrated a Grem1-neutralizing antibody was effective in treating a mouse model of pulmonary hypertension [45]. In the studies presented here, we demonstrated for the first time that the use of a Grem1-neutralizing antibody (UCB Pharma) in the 5TGM1/KaLwRij mouse model of MM resulted in a reduction of up to 81.2% in mean MM tumor burden. Given that the Grem1-neutralizing antibody had no significant effect on the growth of 5TGM1 MM PCs in monoculture in vitro, the effects of the Grem1-neutralizing antibody in vivo are consistent with targeting a PC-extrinsic factor. Despite a recently published study by Rowan et al. that determined that Grem1 depletion in vivo resulted in severe enteropathy and rapid onset BM failure in adult mice [46], the data from this study and a previous study by Ciuclan et al. have demonstrated that anti-Grem1 antibody treatment is well tolerated, with no adverse effects reported in any experimental animals [45]. This is in keeping with the favorable safety profile that has been observed in monoclonal antibody therapies compared to standard chemotherapeutics [47].

Importantly, while tumor burden was reduced by more than 5-fold with anti-Grem1 neutralizing antibody treatment, complete eradication of tumor burden was not achieved. Preliminary in vitro studies demonstrated anti-Grem1 antibody treatment resulted in only a limited reduction in MM PC growth when added to MM PC stromal co-cultures and, therefore, that MM PCs do not appear to be completely dependent on Grem1 for their growth and survival. Furthermore, since the studies presented here demonstrated a positive feedback loop whereby MM PC upregulated stromal-derived *Grem1*, it is conceivable that the elevated levels of BM *Grem1* cannot be effectively neutralized by the administration of a single anti-Grem1 agent. This may also explain the greater antitumor effect observed when treatment was commenced prior to tumor cell inoculation (81.2% reduction) compared to when treatment was started posttumor cell inoculation (54.5% reduction). In the latter situation, the tumor cells have longer to interact with the BM stromal cells, initiating this positive feedback loop within the tumor microenvironment. As such, a combination therapy approach that utilizes current frontline therapies targeting the MM PCs and their interaction with the cellular compartment of the BM microenvironment may achieve a more complete response. In support of this notion, previous clinical studies examining the effectiveness of the monoclonal antibody against the MM PC cell-surface protein SLAMF7 (elotuzumab) showed limited clinical activity as a single agent, but was able to limit MM disease progression and improve overall survival when combined with frontline MM therapies [48]. Furthermore, disease relapse is often inevitable in MM and patients standardly receive a period of maintenance therapy after their initial induction treatment regimen. Given that anti-Grem1 antibody treatment demonstrated the greatest antitumor effect during early stages of disease, where tumor burden was still limited, this therapy would be well suited in the maintenance phase of treatment where MM PC tumors may still be widespread, but of small size.

Current treatment strategies primarily target the proliferative MM PC population, yet overlook the critical role of the BM microenvironment in supporting MM PC growth, survival, and drug resistance [49]. While this is an area of great research interest, there has been limited success and very few therapies targeting the MM BM microenvironment that have made it into standard clinical practice [49]. There remains a need to identify new BM therapeutic targets and devise new strategies to effectively reduce MM tumor burden. Collectively, our data suggest that Grem1 is a key BM stromal-derived factor that promotes MM disease initiation and progression and that antibody-mediated targeting of Grem1 significantly reduces disease burden. In summary, our findings suggest that an anti-Grem1 antibody therapy warrants further investigation in MM, with the potential to achieve a sustained antitumor response in the upfront setting, and may be ideally suited for inclusion as a maintenance therapy to prevent and/or delay disease relapse.

## 4. Materials and Methods

### 4.1. Cell Culture

All cell lines were maintained in a humidified environment at 37 °C with 5% carbon dioxide. Unless otherwise stated, all cell culture reagents were sourced from Sigma-Aldrich (St. Louis, MI, USA) and all media were supplemented with 10% (*v/v*) fetal calf serum (FCS), 2 mM L-glutamine, 1 mM sodium pyruvate, 15 mM 4-(2-hydroxyethyl)-1-piperazineethanesulfonic acid (HEPES) buffer, 100 U/mL penicillin, and 100 μg/mL streptomycin. The murine MM 5TGM1 PC line was kindly provided by Assoc. Prof. Claire Edwards (University of Oxford, Oxford, UK). The 5TGM1 cells expressing both green fluorescent protein (GFP) and firefly luciferase were generated using the retroviral expression vector NES-TGL [48], and a new clonal subline that exhibits consistent bone tropism (5TGM1.Bmx1) was established [48]. The 5TGM1 cells were maintained in Iscove’s Modified Eagle’s Medium. The mouse bone marrow stromal cell line OP9 was obtained from the American Type Culture Collection (Manassas, VA, USA) and maintained in Dulbecco’s Modified Eagle Medium (DMEM). Co-cultures of 5TGM1 cells and OP9 cells were maintained in Iscove’s Modified Eagle’s Medium (IMDM). The human MM cell lines (HMCLs) RPMI-8226, U266, and KMS-11 were obtained from the ATCC. The HMCL H929 was kindly provided by Prof. Andrew Spencer (Monash University, Melbourne, Australia). All HMCLs were cultured in RPMI-1640 medium. The MDA-MB-231-TXSA breast cancer cell line was kindly provided by Dr. Toshiyuki Yoneda (formerly at University of Texas Health Sciences Center, San Antonio, TX, USA) and maintained in RPMI-1640 medium.

### 4.2. Human Bone Marrow Stromal Cell Purification and RNA Isolation

Iliac crest trephines were collected from randomly selected patients with symptomatic MM who presented at the Royal Adelaide Hospital (Adelaide, Australia) and from hematologically normal age-matched controls (MM; *n* = 15, Normal; *n* = 17, age-range = 44–78 years, mean age = 61.9 years). All patients provided informed consent in accordance with the Declaration of Helsinki. Bone marrow (BM) mononuclear cells were prepared from BM trephines by density gradient isolation, as previously described [50], and cryopreserved by the South Australian Cancer Research Biobank at SA Pathology. The studies were approved by the Central Adelaide Local Health Network Human Research Ethics Committee (HREC/13/RAH/569 No:131133). Samples were collected from patients prior to treatment. Stromal cell cultures were grown out ex vivo from the BM mononuclear cells by plastic adherence culture as previously described [1] and expanded prior to cryopreservation. Stromal cell cultures were retrieved from storage in liquid nitrogen and cultured ex vivo at passage 2 in alpha-MEM culture medium supplemented with 100 µM L-ascorbate-2-phosphate for 24 h prior to RNA extraction using TRIzol^TM^ Reagent (Thermo Fisher Scientific, Waltham, MA, USA).

### 4.3. Murine Compact Bone Purification and RNA Isolation

C57BL6/KaLwRij.Hsd (KaLwRij) mice were kindly provided by Prof. Andrew Spencer (Monash University, Melbourne, Australia) and were rederived, bred, and housed at the South Australian Health and Medical Research Institute (SAHMRI) Bioresources Facility. Vk*MYC 4929 splenic-derived transplants were obtained from Prof. Ricky Johnstone (Peter MacCallum Cancer Centre, Melbourne, Australia) and passaged through C57BL6 mice. All procedures were performed with the approval of the SAHMRI Animal Ethics Committee (Ethics Approval #SAM165 and #166). The 6–8-week-old KaLwRij or C57BL6 mice were injected intravenously (i.v.) with 5 × 10^5^ 5TGM1.Bmx1 MM PCs and 1 × 10^6^ Vk*MYC 4929 splenic transplant cells, respectively, in 100 µL of sterile phosphate buffered saline (PBS). For 5TGM1-tumor bearing KaLwRij mice, tumor growth was established over four weeks (as previously described [32]). Tumor development at the study endpoint was confirmed by in vivo whole animal bioluminescent imaging (BLI) using a Xenogen IVIS Spectrum Imaging System (Perkin Elmer Inc., Waltham, MA, USA) after intraperitoneal injection (i.p.) of 150 mg/kg of D-Luciferin (Biosynth, Basel, Switzerland). Tumor burden was quantitated using Living Image software (Perkin Elmer Inc). Vk*MYC tumor growth was established over 12 weeks and validated by presence of an M-spike in peripheral blood serum. At the experimental endpoint, femora and tibiae from tumor-bearing, and age- and sex-matched nontumor mice were isolated, and the BM flushed out with PFE buffer (PBS, 2% FCS, and 2 mM Ethylenediaminetetraacetic acid (EDTA)). Collagenase digestion of the cortical and trabecular bone was performed, as previously described [51]. The digested bone fragments and collagenase-isolated BM stromal cells were co-lyzed in 1 mL of TRIzol^TM^ Reagent (Thermo Fisher Scientific) and incubated on ice for 15 min.

### 4.4. Quantitative Reverse Transcription Polymerase Chain Reaction (RT-qPCR)

Total RNA was isolated from cells using TRIzol^TM^ Reagent (Thermo Fisher Scientific) according to the manufacturer’s instructions, unless otherwise specified. All RNA samples underwent DNase treatment with RQ1 DNase (Promega, Madison, WI, USA), as per manufacturer’s instructions. RNA (1 µg) was reverse transcribed into complementary DNA using SuperScriptIII^TM^ (Thermo Fisher Scientific) according to manufacturer’s instructions. Real-time polymerase chain reaction (PCR) was conducted using 1 × RT2 SYBR^®^ Green qPCR Mastermix (QIAGEN, Hilden, Germany) and the following primer sequences on the CFX Connect^TM^ Real-Time PCR Detection System (Bio-Rad, Hercules, CA, USA): Mouse Grem1: forward 5′-GCGCAAGTATCTGAAGCGAG-3′; reverse 5′-CGGTTGATGATAGTGCGGCT-3′, human GREM1: forward 5′- AGGCCCAGCACAATGACTCAG-3′; reverse 5′- GTCTCGCTTCAGGTATTTGCG-3′, mouse Il-6; forward 5′-GCACTCCTTGGATAGAGCCC-3′; reverse 5′- ACGAGGATTCTTGCACTGGG-3′, human IL-6; forward 5′-CCAGTACCAATGCGTCATCCA-3′; reverse 5′-CTGGGCTCTGCTATCCAAGGAG-3′; mouse/human β-actin: forward 5′- GATCATTGCTCCTCCTGAGC; reverse 5′-GTCATAGTCCGCCTAGAAGCAT-3′. Changes in gene expression were calculated relative to β-actin as the endogenous control using the standard curve method.

### 4.5. Generation of a Murine Grem1-Overexpressing Stromal Cell Line

An OP9 BM stromal cell line, constitutively overexpressing Grem1, was generated by infection with a pLegoiT2 lentiviral vector (Plasmid #27343, Addgene) harboring the murine cDNA for the coding region of Grem1, isolated from a pCMV6-Kanamycin resistant vector kindly provided by Dr. Miao Yang (Gastrointestinal Cancer Biology Group, SAHMRI, Adelaide, Australia). The pLegoiT2-Grem1 viral vector was transfected into HEK-293T cells using Lipofectamine2000^TM^ (Thermo Fisher Scientific) together with packaging plasmids psPAX2 and EcoENV, and viral particle-containing supernatant was used to infect OP9 stromal cells. Cells were sorted for expression of tdTomato fluorescent protein on a FACSAria^TM^ Fusion (BD Biosciences, San Jose, CA, USA). Transgene expression was confirmed by RT-qPCR and Western blot, as previously described [31].

### 4.6. Western Blot Analysis

Briefly, protein extracted from cell lysates by radioimmunoprecipitation assay buffer (RIPA) lysis buffer [31] was separated on a 10% acrylamide gel and subjected to sodium dodecyl sulphate-polyacrylamide gel electrophoresis (SDS-PAGE). Proteins were transferred to a nitrocellulose membrane and subsequently blocked with 5% *w/v* skimmed milk powder. Immunoblotting was performed with primary antibodies directed against Grem1 (clone 140010, Abcam, 1:250), phospho-Smad1/5/9 antibodies (D5B10, Cell Signaling Technologies, Danvers, MA, USA, 1:1000 dilution). Hsp90 (cat# 7942, Santa Cruz, Dallas, TX, USA, 1:2500) and β-actin (cat# A1978, Sigma Aldrich, 1:2500 dilution) antibodies were used as loading controls. Blots were incubated with an alkaline phosphatase (AP)-conjugated anti-rabbit secondary antibody (cat# 12-448, Millipore, Burlington, MA, USA, 1:2500 dilution). Blots were subjected to an ECF^TM^ substrate prior to visualization of proteins on the ChemiDoc Imaging System (Bio-Rad). Uncropped blots can be found at Appendix A.

### 4.7. Co-Culture of MM PC and BM Stroma

OP9 stromal cells were seeded (contact culture; 5 × 10^4^ cells per 6-cm dish, Transwell culture; 2 × 10^4^ per well in a 24-well plate) in triplicate and allowed to adhere for 5 h. The 5TGM1 MM PC were added to the stromal cell cultures at 1 × 10^5^ cells/mL (contact culture; 5 mL, Transwell culture; 100 µL in Transwell insert). For contact co-culture, GFP-positive OP9 cells were isolated from GFP-negative 5TGM1 parental cells by fluorescent activated cell sorting on a FACSAria^TM^ Fusion (BD Biosciences). Stromal cells were collected in TRIzol^TM^ Reagent at 24, 48, and 72 h post co-culture initiation. Human MM cell lines RPMI-8226, U266, KMS-11, and H929 were cultured with three primary human BM stromal samples isolated from hematopoietically healthy individuals for 72 h. Human MM cell lines were washed thoroughly from the adherent stroma twice with 1 × PBS prior to collection of stromal cells in TRIzol^TM^ Reagent. For Il-6 experiments, cells were cultured in 24-well plates for 72 h as per the above co-culture conditions. The 20-ng/mL recombinant mouse Il-6 protein (cat #I9646-5UG, Sigma Aldrich) and 0.05 µg/mL neutralizing anti-mouse Il-6 antibody (cat #AF-406-NA, R&D systems, Minneapolis, MN, USA) were used in cultures as specified.

### 4.8. Luciferase Proliferation Assay

The 5TGM1 MM PC line was cultured with either Grem1-overexpressing or empty vector OP9 stromal cells for 72 h in a 24-well plate, as described above. Following incubation, cells were collected and lysed in 1 × Passive Cell Lysis Buffer (Promega). The 20 µL of cell lysate was transferred to a 96-well plate. Immediately prior to reading the plate, 100 µL of luciferase reaction buffer (5 mM MgCl_2_, 30 mM HEPES, 150 μM ATP, 500 µg/mL of Coenzyme A, and 150 μg/mL D-luciferin) was added to the cell lysate. Luminescence was measured using a Wallac 1420 Victor Microplate reader (Perkin Elmer), with luminous intensity used as a direct measure of MM PC number.

### 4.9. HEKID1 Reporter Assay

Clone 12 cells were cultured in DMEM containing 10% FCS, 1 × L-Glutamine, 1 × non-essential amino acids, and Hygromycin B (200 µg/mL) to ensure cells do not lose Id1 gene expression. Cell were assayed in DMEM containing 0.5% FCS, 1 × L-Glutamine, and 1 × NEAA. Cells were seeded in 96-well Poly-D-Lysine-coated plate at 5 × 10^4^ cells per well and incubated for 3-4 h prior to assay. Then, 10 µg/mL BMP heterodimer stock (UCB Pharma, Brussels, Belgium) was prepared at 100× and added to cells. Then, 0.2 nM of recombinant mouse Grem1 protein (UCB Pharma) was added to cells. Three-fold titrations of anti-Grem1 antibody (UCB Pharma) was added to cells, to a maximum dose of 15 nM. All wells were made up to 60 µL with assay medium and incubated for a further 45 min at 37 °C. Post-incubation, 30 µL of sample was transferred per well of assay plate and incubated for 20–24 h before measuring luminescence signal. Cell Steady Glo (Promega) was added to assay plates at room temperature. Luciferase signal was detected by addition of Cell Steady Glo reagent (100 µL) for 20 min on shaker at room temperature and measuring luminescence using Cell Titre Glo protocol (Promega) on Synergy 2 (BioTek, Winooski, VT, USA).

### 4.10. Confirmation of Anti-Grem1 Antibody Activity

MDA-MB-231-TSXA cells were seeded into 6-well plates and cultured until 80% confluent. Cells were starved in serum-free medium overnight, and then stimulated with recombinant Grem1 (UCB Pharma) and/or recombinant BMP2 (ProSpec, Rehovot, Israel) as indicated for 2 h. Cell lysates were collected for analysis by Western blot as previously described.

### 4.11. Targeting Grem1 in an Immunocompetent Murine Model of Systemic MM

Grem1-neutralizing antibody was kindly provided by Dr. Gareth Davies (UCB Pharma). Age- and sex-matched KaLwRij mice (6–8 weeks old) were assigned to one of three treatment regimens with 30 mg/kg Grem1-neutralizing antibody or IgG control (UCB Pharma) by subcutaneous (s.c.) injection. In pretumor (treatment started one week prior to tumor cell inoculation) and three days posttumor cell inoculation treatment regimens, mice were treated twice weekly from the first antibody injection for the duration of the 4-week model. In the late-stage disease model, mice received the first dose of Grem1-neutralizing antibody or IgG control two weeks posttumor cell inoculation, and a second, final dose one week later. Briefly, KaLwRij mice were inoculated with 5 × 10^5^ bone tropic, GFP, and luciferase-positive 5TGM1 cells (5TGM1.Bmx1) via tail vein injection. Tumor development was monitored weekly by in vivo BLI, as described earlier. At experimental endpoints peripheral blood serum was isolated and serum protein electrophoresis was performed using the Hydragel30 β1β2 kit (Sebia) according to the instructions of the manufacturer. The intensity of paraprotein band/M-spike was quantitated relative to the albumin band using Image Lab Software v6.0.1 (Bio-Rad).

### 4.12. Cell Lineage Flow Cytometric Analysis

For flow cytometric analysis, long bones (femora and tibiae) were excised and cleaned. Bones were gently crushed with a mortar and pestle in PFE to isolate BM and compact bone cells were isolated, as previously described [51,52]. Briefly, the resultant bone chips were washed, finely chopped, and incubated with 3 mg/mL collagenase type-I (ScimaR, VIC, Australia) in PBS at 37 °C for 45 min. Both cell suspensions were strained through a 70-µm cell strainer, resuspended in PBS, and stained with Fixable Viability Stain 700 (323 ng/mL; BD Biosciences) for 15 min. Cells were washed, resuspended at 1 × 10^7^c ells/mL in PFE, and blocked with mouse gamma globulin (117 μg/mL; Abacus ALS, QLD, Australia) for 30 min on ice.

For analysis of hematopoietic progenitor cells, a lineage cocktail of biotin-conjugated antibodies (B220, CD3, CD4, CD5, CD8, Gr1, Ter119 (BioLegend, San Diego, CA, USA) and Cd11b (eBioscience, San Diego, CA, USA)) stained with streptavidin-APC (Life Technologies, Carlsbad, CA, USA) secondarily for 30 min was used to exclude mature Lin+ cells. Hematopoietic progenitor cells were concurrently stained with Sca-Brilliant Violet-(BV)786, cKit-PE-Cy7, CD135-PE-CF594, and CD34-BV421 (all from BD Biosciences). Endothelial lineage cells were concurrently stained with CD31-PerCP/Cy5.5 and CD44-PE (all from BD Biosciences). Mesenchymal lineage cells (MSC, osteoprogenitors, and mature osteoblasts) were quantitated from compact bone preparation, as previously described [52]. All antibody cocktails were comprised of rat anti-mouse antibodies. Finally, cells were washed, filtered, and fixed in 1% neutral buffered formalin, 2% glucose, and 0.01% sodium azide in PBS prior to analyzing on the LRSFortessa X20 (BD Biosciences). All flow cytometric data was subsequently analyzed using Flowjo v10 with cell populations of interest analyzed as a percentage of total viable cells or as a percentage of the parent population.

### 4.13. Statistics

All statistical analyses were performed using GraphPad Prism v.8.0.0 (GraphPad Software). For in vitro experiments, the comparison of two groups for a single variable used a parametric paired or unpaired *t*-test and was presented as mean ± SEM for three independent replicate experiments. For in vivo experiments or data that were not normally distributed, the comparison of two groups for a single variable used a nonparametric Mann–Whitney *U* test and was presented as median ± interquartile range for individual samples analyzed. When three or more groups were compared for a single variable, a one-way analysis of variance (ANOVA) with Tukey multiple comparisons was used. For time-course experiments, groups were compared using a two-way ANOVA with Sidak’s multiple comparisons test. Correlations were assessed using Pearson correlation coefficients. Differences were considered statistically significant when *p* < 0.05.

## 5. Conclusions

This study suggests that Grem1 is a key stromal-derived factor that promotes MM disease, and that antibody-mediated targeting of Grem1 significantly reduces disease burden. With few effective therapies that target the critical relationship between MM PCs and the BM, the findings presented here represent a novel treatment strategy to limit MM disease burden.

## 6. Patents

Zannettino A.C.W., Hewett D., Clark K., & Panagopoulos V. (2019). Prevention and treatment of cancer. UK provisional patent number GB1809946.5. University of Adelaide, UCB Pharma SPRL, and The Oxford University Innovation Limited. Patent filed June 2019.

## Figures and Tables

**Figure 1 cancers-12-02149-f001:**
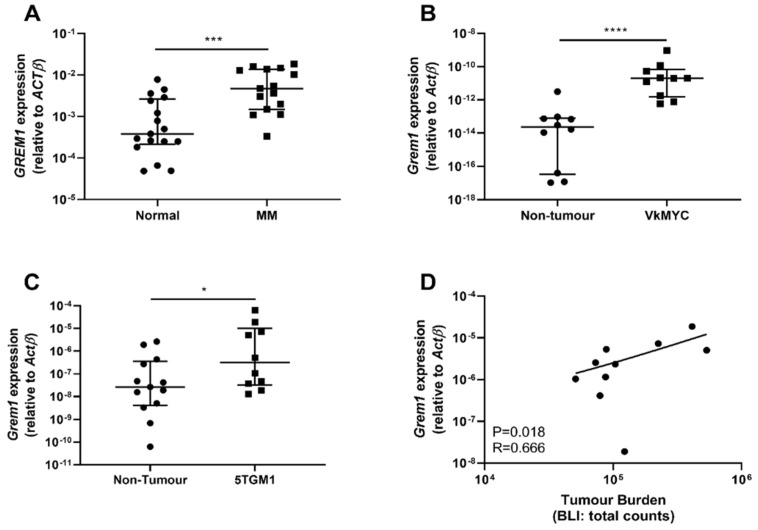
*GREMLIN1 (GREM1)* expression is elevated in primary stromal cultures from multiple myeloma (MM) patients and the compact bone of tumor-bearing KaLwRij mice. RNA was extracted from stromal cells ex vivo cultured from bone marrow (BM) trephine samples from age- and gender-matched normal donors and MM patients and the expression of *GREM1* (**A**) was analyzed by real-time PCR. Data presented as mRNA expression normalized to β-Actin (*ACTB*), median ± interquartile range, Normal; *n* = 17 and MM; *n* = 15, *** *p* < 0.001, Mann–Whitney test. *Grem1* expression in compact bone isolated from the long bones of nontumor- and (**B**) Vk*MYC tumor-bearing C57BL6 mice (*n* = 10), and (**C**) 5TGM1 tumor-bearing (MM) KaLwRij mice (Nontumor; *n =* 13 and Tumor-bearing; *n =* 10) was analyzed by real-time PCR. Graph depicts *Grem1* expression relative to β-Actin (*ActB*), median ± interquartile range, * *p* <0.05, **** *p* < 0.0001, Mann–Whitney test. (**D**) *Grem1* expression in the BM stroma isolated from the compact bone of the hindlimbs of KaLwRij tumor-bearing mice was correlated with the tumor burden in the respective limbs, as detected by bioluminescent imaging (BLI). Graph depicts *Grem1* expression (*y*-axis) vs. tumor burden (*x*-axis), *n =* 10, Pearson correlation, *p* < 0.05, r = 0.666.

**Figure 2 cancers-12-02149-f002:**
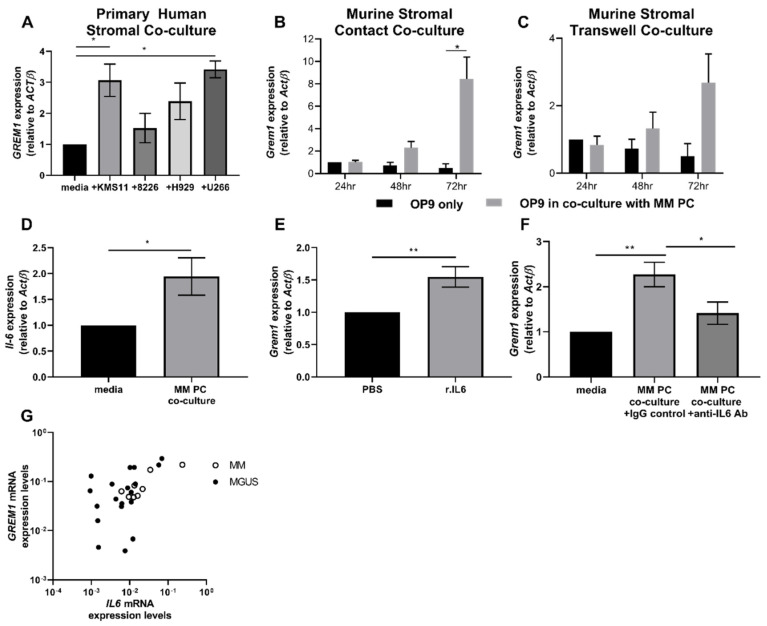
MM plasma cells (PCs) upregulate stromal expression of *Grem1* in adherent co-culture via interleukin-6 (Il-6). RNA was extracted from (**A**) washed primary, normal human stroma previously co-cultured with KMS-11, RPMI-8226, H929, and U266 human MM cell lines for 72 h and (**B**) fluorescent-activated cell sorting (FACS) sorted green-fluorescent protein (GFP)+ murine, bone-marrow-derived stromal OP9 cell lines previously co-cultured with 5TGM1 MM PCs in cell-contact and (**C**) Transwell culture conditions for 24, 48, and 72 h. *GREM1/Grem1* expression from stromal samples was analyzed by qPCR. (**D**) *Il-6* expression was analyzed in the murine, BM-derived, stromal OP9 cell lines following 72 h culture in the absence/presence of 5TGM1 MM PCs. (**E**) OP9 stromal cells were cultured in the absence/presence of 20 ng/mL murine recombinant Il-6 (r.Il-6) for 72 h and analyzed for *Grem1* expression. (**F**) OP9 stromal cells were cultured in either media only or in co-culture with 5TGM1 MM PCs in presence of 0.05 µg/mL anti-Il-6 neutralizing antibody (MM PC co-culture + anti-Il-6 Ab) or immunoglobulin G (IgG) antibody control (MM PC co-culture + IgG control) and analyzed for *Grem1* expression. Graphs depict data presented as mRNA expression normalized to β-Actin (*ACTB/ActB*), mean ± standard error of the mean of three replicate experiments; * *p* < 0.05, ** *p* < 0.01, *** *p* < 0.001, (**A**–**C**,**F**) one-way ANOVA, Tukey multiple comparisons and (**D**,**E**) *t*-test. (**G**) *GREM1* mRNA expression from ex vivo cultured monoclonal gammopathy of undetermined significance (MGUS) and MM patient BM stromal samples was correlated with *IL-6* mRNA expression. Graph depicts *GREM1* expression (*y*-axis) vs. *IL6* expression (*x*-axis), MGUS; *n =* 21 and MM; *n* =8, Pearson correlation, MGUS; *p* < 0.01, *r* = 0.533 and MM; *p* < 0.05, *r* = 0.738.

**Figure 3 cancers-12-02149-f003:**
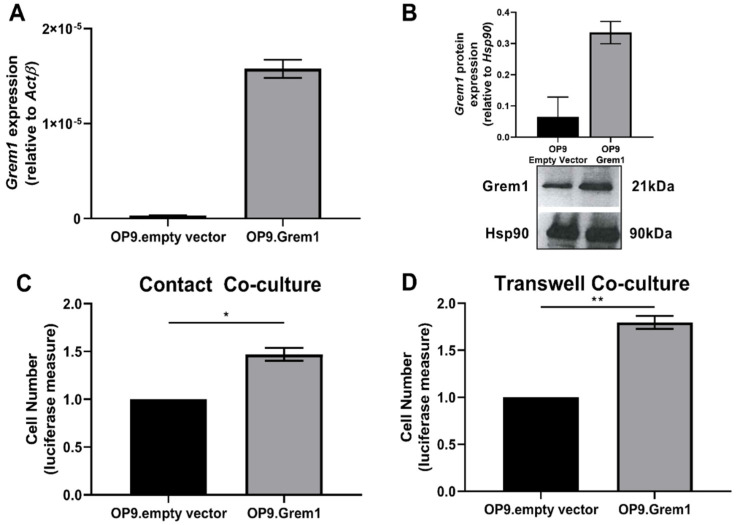
Elevating BM stromal cell expression of Grem1 promotes an increase in 5TGM1 MM PC proliferation. *Grem1* was overexpressed in the murine BM stromal cell line, OP9. Grem1 transgene expression in OP9 stromal cells was confirmed by (**A**) RT-PCR and (**B**) representative Western blot (graph depicting densitometric analysis of two independent Western blots). Proliferation of 5TGM1 MM PCs in (**C**) cell-cell contact and (**D**) Transwell co-cultures with OP9 empty vector and Grem1-overexpressing OP9 (OP9.Grem1) stromal cells was measured by luciferase activity after 72 h of culture. Graphs depict mean ± SEM of three replicate experiments normalized to the OP9 empty vector control, * *p* < 0.05, ** *p* < 0.01, *t*-test.

**Figure 4 cancers-12-02149-f004:**
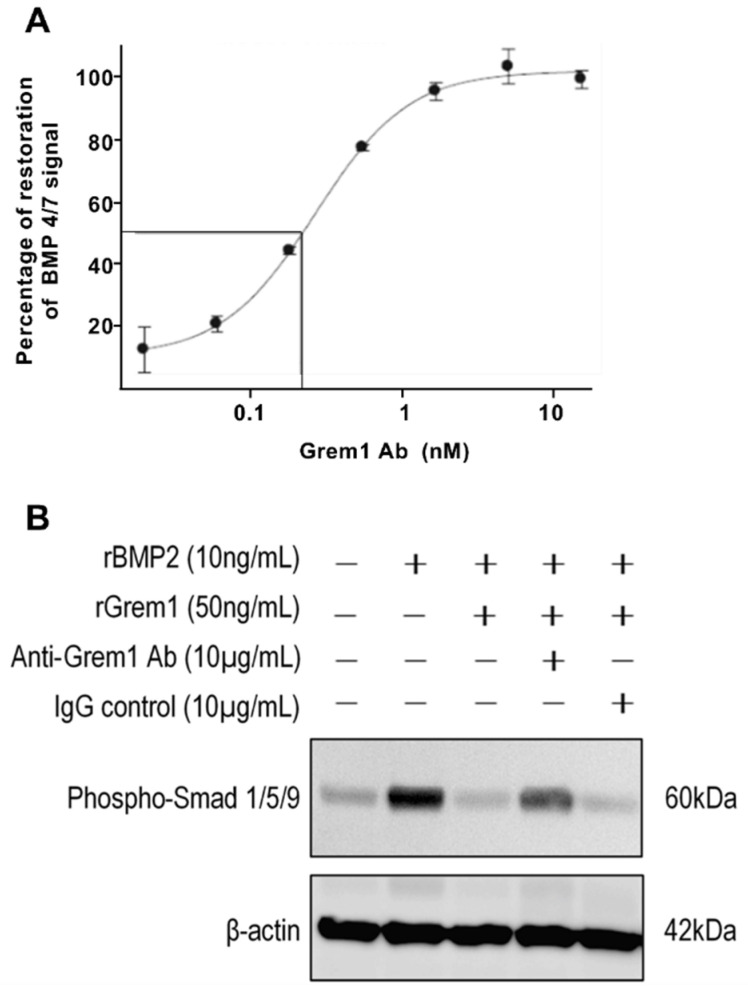
Anti-Grem1 antibody nullifies Grem1 inhibition of bone morphogenic protein (BMP) signaling. (**A**) Clone 12 cells were cultured with serial dilutions of anti-Grem1 antibody protein in the presence of a fixed amount of recombinant Grem1 protein and BMP heterodimer protein as part of a human embryonic kidney (HEK) Inhibitor of Binding Protein-1 (ID1) reporter assay. Data shown as percentage restoration of BMP 4/7 signal as measured by luciferase signal, relative to BMP4/7 signal in the absence of Grem1. (**B**) MDA-MB-231-TXSA cells were serum starved for 6 h, before stimulation for 2 h with combinations of recombinant BMP2 (rBMP2), recombinant Grem1 (rGrem1), anti-Grem1 antibody, and IgG isotype control antibody. Protein was immediately isolated from cells and analyzed by Western blot. BMP signaling was indicated by the amount of phospho- Mothers against decapentaplegic homolog (Smad) 1/5/9 protein relative to the loading control β-actin.

**Figure 5 cancers-12-02149-f005:**
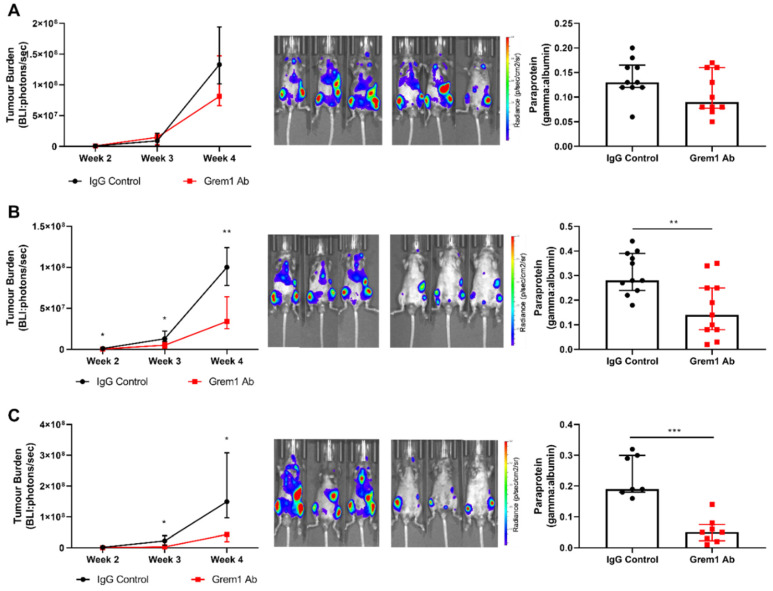
Anti-Grem1 antibody treatment reduces 5TGM1 tumor growth in vivo. KaLwRij mice were inoculated with 5 × 10^5^ 5TGM1.Bmx1 MM PCs. Mice were treated with either 30 mg/kg anti-Grem1 antibody or IgG isotype control antibody in three treatment regimens: (**A**) Late-stage established disease (once weekly treatment administered at weeks 2 and 3 posttumor cell inoculation, *n* =10/treatment group), (**B**) early-stage established disease (twice weekly treatment administered from day 3 posttumor cell inoculation, *n* =11/treatment group), and (**C**) prior to tumor engraftment (twice weekly treatment administered one week prior to tumor cell inoculation, *n* = 7–8/treatment group). Tumor burden was measured throughout the study by bioluminescent imaging (BLI) and at the endpoint by serum protein electrophoresis (SPEP). Line graphs depict whole-body ventral BLI quantitation at weeks 2, 3, and 4 for anti-Grem1 antibody- and isotype control antibody-treated mice, two-way ANOVA, Sidak’s multiple comparisons, median ± interquartile range. Representative BLI images for the final timepoint at week 4 are shown. A secondary, independent measure of tumor burden was also used. Graph depicts level of paraprotein detected in serum following tail bleeds at week 4 (normalized to internal albumin control), Mann–Whitney test. Scatterplots depict median ± interquartile range, Mann–Whitney test, of *n* = 11 mice per treatment group, * *p* < 0.05, ** *p* < 0.01, *** *p* < 0.001.

**Figure 6 cancers-12-02149-f006:**
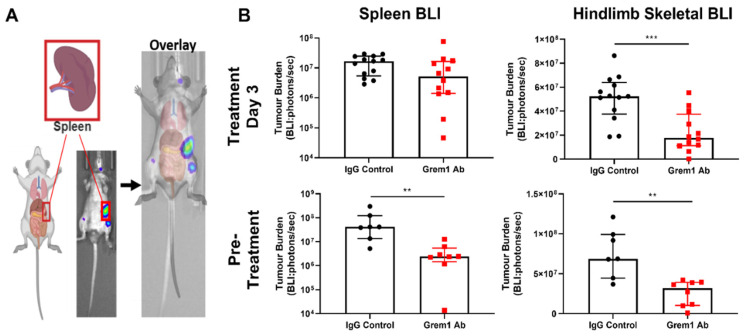
Anti-Grem1 antibody treatment in 5TGM1 tumor-bearing mice specifically reduces skeletal tumor burden, but not splenic tumor burden. (**A**) Schematic illustrating the detection of splenic tumor signal by BLI. (**B**) Final BLI scans for anti-Grem1 treatment regimens started either three days posttumor cell inoculation (posttreatment) or prior to 5TGM1 tumor cell inoculation (pretreatment) measuring BLI tumor signal from the splenic region and skeletal BLI of the long bones. Graphs depict median ± interquartile range, posttreatment spleen: *n* = 13; posttreatment long bones: *n* = 13; pretreatment spleen: *n* = 7; and pretreatment long bones: *n* = 8. ** *p* < 0.01, *** *p* < 0.001, Mann–Whitney test.

**Figure 7 cancers-12-02149-f007:**
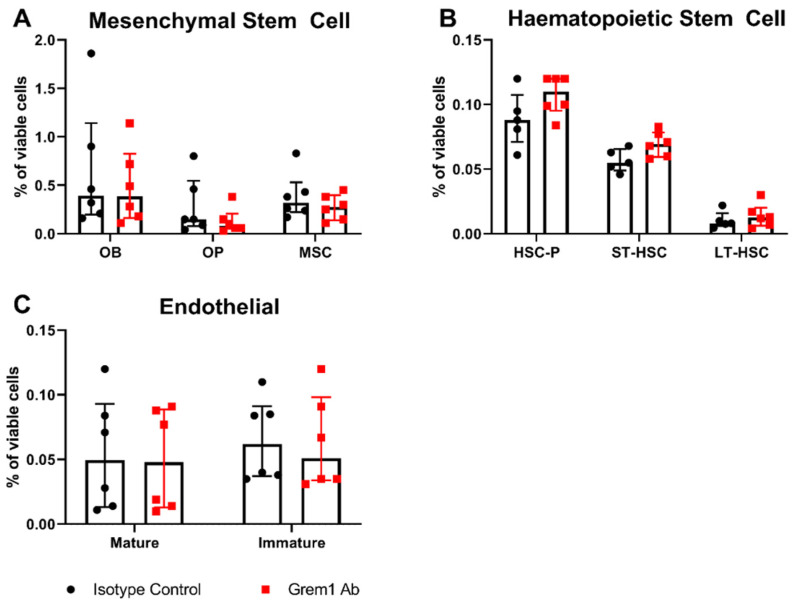
Anti-Grem1 antibody treatment does not alter the cellular composition of the BM in vivo. KaLwRij mice were treated with 30 mg/kg anti-Grem1 antibody or IgG control antibody twice weekly for two weeks. Mice were culled three days after the final treatment and BM and compact bones were collected from the long bones for flow cytometric analysis. Compact bone was analyzed for (**A**) MSC lineage populations: OB (CD45−, Lin−, CD31−, CD51+, Sca1+), OP (CD45−, Lin−, CD31−, CD51+, Sca1−), and MSCs (CD45−, Lin−, CD31−, CD51−, Sca1+). BM was also analyzed for: (**B**) HSC lineage populations: Progenitors (Lin−, Sca1+, cKit+), short-term HSC (Lin−, Sca1+, cKit+, CD135−, CD34+), long-term HSC (Lin−, Sca1+, cKit+, CD135−, CD34−), and (**C**) mature (CD31+CD144+) and immature (CD31+CD144−) endothelial lineages. Graphs depict median ± interquartile range from *n* = 6 mice per treatment group from two independent experiments; *p* > 0.05, unpaired *t*-test. OB = osteoblast, OP = osteoprogenitor, MSC = mesenchymal stem cell, HSC-p = hematopoietic stem cell progenitors, ST-HS C = short-term hematopoietic stem cell, LT-HSC = long-term hematopoietic stem cell.

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
