# Peer review of "Targeted Disruption of Bone Marrow Stromal Cell-Derived Gremlin1 Limits Multiple Myeloma Disease Progression In Vivo"

_cancers, 2020, doi:10.3390/cancers12082149_

Round 1

Reviewer 1 Report

The authors suggest  Gremlin1 (Grem1) plays a critical role in MM disease development, with interaction with the stromal cells and myeloma. The study is well designed and the manuscript is well written. There are some questions to be answered from the manuscript.

  1. Since IL-6 is the major player for myeloma proliferation, the authors should have more studies about IL-6 and Grem1 for a clear picture of their interaction.
  2. The study suggests that Grem1 is a key BM stromal-derived factor that promotes MM disease initiation and progression, but why the author state that anti-Grem1 therapy might be suitable for maintain therapy, not the initial treatment in the discussion and summary section.
  3. Why the response of anti-Grem1 therapy have organ-specific effects with different  response between different organ (spleen vs bone)?

Reviewer 2 Report

In this paper authors focus, for the first time, on Gremlin1, a highly conserved protein with a critical role in multiple myeloma (MM) development in vitro and in vivo.

The topic is of considerable interest and the papers is well organized and written.

Minor Remarks:

- In the Results section authors state (line 163-165): “OP9 stromal cells overexpressing Grem1 were generated (Figure 3A-B) and used in co-cultures with 5TGM1 MM PCs.” The authors should reveal the strategy to obtain OP9 overexpressing Gremlin1 and should better explain the Figure 2B to simplify the comprehension of performed experiments.

- In the Results section authors show that MM PCs upregulate stromal expression of Gremlin1 in adherent co-culture via IL-6. Since in the Discussion section authors state (line 324-327): “Paracrine and autocrine upregulation of IL-6 within the MM BM microenvironment is a common feature of MM and it is well understood that IL-6 promotes the survival and proliferation of MM tumour cells, mediated through JAK/STAT upregulation of VEGF, Ras, Akt and MAPK pathways.”, they should perform experiments to demonstrate the role of Gremlin1 in the regulation of these pathways.

- It is recommended that the authors also reveal the role of Gremlin1 in MGUS patients to understand its role in the disease progression.

- In the Materials and Methods section the sources of different companies are missing. Authors should add them.

- In the text, there are typing mistakes and often abbreviations are missing. Authors should resolve in order to submit a neat paper.

Reviewer 3 Report

In this paper Clark and coauthors propose a new interesting mechanism explaining the pathogenesis of multiple myeloma by showing that Gremlin1 is overexpressed by stromal cells of patients and diseased mice. Of note, they demonstrate that plasma cells can induce this overexpression, and that the treatment with Gremlin1 neutralizing antibodies can counteract the progression of the disease. The paper is well written and the study design is very convincing and well conducted.

Some major concerns are as follow:

  • It is not clear why the authors chose to use mean±SD for some data and mean±SEM for others. This must be clearly stated. For example in Figure 1, do dots represent single samples or are they repetitions of same samples? If they are related to different samples, authors must use mean±SD and repeat the statistical analysis. If not this must be stated. The same for all the other graphs.
  • Authors must clearly state the passage of the ex vivo culture of stromal cells. It is known that this could affect gene expression even in physiologic samples. Did they compare early passages of normal and diseased samples? Further, it would be nice to add some morphologic/functional data about these cells: i.g. what about cell morphology and proliferation rate?
  • It is not clear how the authors could select stromal cells after direct co-culture for molecular analysis. How the authors excluded the presence of the contaminating RNA from the co-cultured cells? I would suggest to add the basal expression of Gremilin1 in parallel samples from single cell cultures of stromal cells and of all the cell lines at least as a supplement.
  • In Figure 2F, it would be useful to add data from single cell culture plus Ig or neutralizing antibodies? Can the author appreciate any effect as well?
  • In the western blot from Figure 3B, Hsp90 seems to increase according to Gremilin1. Authors must show densitometric analysis of at least three single experiments. Also molecular expression should be confirmed in more than one sample.
  • The analysis of bone marrow composition in vivo should be accompanied by an histological view.

Minor:

  • The manuscript is well written, but sometimes not well formatted. Further, the whole manuscript presents to many acronyms that make it difficult a fluent reading. I would suggest to remove the most of them and keep just some.
  • Graphs are poorly presented. I suggest to conform all to a single common style.

Round 2

Reviewer 3 Report

Authors adequately addressed all my concerns.